# Whole-Genome Omics Elucidates the Role of CCM1 and Progesterone in Cerebral Cavernous Malformations within CmPn Networks

**DOI:** 10.3390/diagnostics14171895

**Published:** 2024-08-28

**Authors:** Jacob Croft, Brian Grajeda, Liyuan Gao, Johnathan Abou-Fadel, Ahmed Badr, Victor Sheng, Jun Zhang

**Affiliations:** 1Departs of Molecular & Translational Medicine (MTM), Texas Tech University Health Science Center El Paso (TTUHSCEP), El Paso, TX 79905, USA; 2Department of Biological Sciences, University of Texas at El Paso, El Paso, TX 79902, USA; 3Department of Computer Sciences, Texas Tech University, Lubbock, TX 79409, USA; 4Department of Anesthesiology, Ochsner LSU Health, Shreveport, LA 71130, USA; aebadr1@gmail.com

**Keywords:** whole-genome omics, cerebral cavernous malformations (CCMs), CCM signaling complex (CSC), blood–brain barrier (BBB), nuclear progesterone receptor (nPRs), non-classic membrane receptor (mPRs), progesterone (PRG) signaling, CSC-mPRs-PRG-nPRs (CmPn) signaling network

## Abstract

Cerebral cavernous malformations (CCMs) are abnormal expansions of brain capillaries that increase the risk of hemorrhagic strokes, with CCM1 mutations responsible for about 50% of familial cases. The disorder can cause irreversible brain damage by compromising the blood–brain barrier (BBB), leading to fatal brain hemorrhages. Studies show that progesterone and its derivatives significantly impact BBB integrity. The three CCM proteins (CCM1, CCM2, and CCM3) form the CCM signaling complex (CSC), linking classic and non-classic progesterone signaling within the CmPn network, which is crucial for maintaining BBB integrity. This study aimed to explore the relationship between CCM1 and key pathways of the CmPn signaling network using three mouse embryonic fibroblast lines (MEFs) with distinct CCM1 expressions. Omics and systems biology analysis investigated CCM1-mediated signaling within the CmPn network. Our findings reveal that CCM1 is essential for regulating cellular processes within progesterone-mediated CmPn/CmP signaling, playing a crucial role in maintaining microvessel integrity. This regulation occurs partly through gene transcription control. The critical role of CCM1 in these processes suggests it could be a promising therapeutic target for CCMs.

## 1. Introduction

Cerebral cavernous malformations (CCMs) are a common type of brain vascular malformation resulting from abnormally dilated brain capillaries. This condition leads to an increased risk of hemorrhagic strokes. CCMs are an autosomal dominant disorder with three known genes, KRIT1 (CCM1), MGC4607 (CCM2), and PDCD10 (CCM3) as causes of familial CCMs [1]. Mutations in the CCM1 gene account for about 50% of familial CCM cases [2,3]. One distinct characteristic of this genetic disorder is its incomplete penetrance, meaning that most people with the mutated gene are asymptomatic. However, when symptoms do occur, they often result in irreversible brain damage. To study the cellular functions of CCM1, researchers have established isogenic mouse embryonic fibroblasts (MEFs) from wild-type (WT) and Ccm1-knockout (KO) mice and created Ccm1-knockin (KI) fibroblasts by stable infection of Ccm1-KO fibroblasts with a lentiviral vector encoding human KRIT1 [4,5,6,7,8].

The blood–brain barrier (BBB) is a crucial interface between the blood and the central nervous system that regulates the flow of substances between the two [9,10]. The BBB integrity is maintained by two major apparatus, adherens junctions (AJs) and tight junctions (TJs), which are formed by different molecules but are functionally and structurally linked [11]. Inflammatory events are a major cause of BBB disruption [12,13], and steroids are often used therapeutically for their anti-inflammatory properties in human conditions, including BBB disorders [11,14,15], although the efficacy of steroids in this context is still a topic of debate [11]. Similar to estrogen and glucocorticoids, the effects of progesterone (PRG) and its derivatives, progestins, on BBB integrity have been investigated, despite a lack of understanding of their off-target effects via their corresponding steroid receptors [16]. It is well established that PRG binds to nuclear receptors (nPR) in classic PRG actions and to membrane progesterone receptors (mPRs) in non-classic PRG actions [17,18,19,20,21,22,23].

Recent findings indicated that CCM2 anchors with both CCM1 and CCM3 to form the CCM signaling complex (CSC) [1,24]. The CSC, in turn, can couple both classic nuclear progesterone receptor (nPRs) and non-classic membrane receptor (mPRs)-mediated progesterone (PRG) signaling to form the CSC-mPRs-PRG-nPRs (CmPn) signaling network in nPR(+) cells, and the CSC-mPRs-PRG (CmP) signaling network in nPR(−) cells [23,25,26]. The CmPn/CmP signaling network was found to play a crucial role in maintaining BBB integrity through its impact on nPR(−) microvascular ECs in both in vitro and in vivo conditions, suggesting the essential roles of CmPn/CmP signaling networks in maintaining microvascular integrity [26,27]. To better understand the cellular roles of CmPn/CmP signaling networks during angiogenesis and tumorigenesis, our team has been conducting multi-omics studies using various cellular and animal models to investigate the key roles of CCM1 within the CmPn signaling network. Specifically, we have focused on the relationship between CCM1 and other key components of the CmPn signaling network (nPRs, mPRs) [23,25]. In this study, we examined the differentially expressed protein (DEP) and gene (DEG) profiles at both the transcriptional and translational levels using mouse embryonic fibroblasts (MEFs) with varying levels of CCM1 protein expression in response to progesterone (PRG) actions. The DEPs were classified into upregulated and downregulated proteins, while the DEGs were categorized into gene-transcribed RNAs for each CCM1 genotype under the influence of progesterone (PRG) and its derivatives.

This comprehensive, unbiased whole-genome analysis enabled us to quantitatively assess genome-wide dynamic changes in response to progesterone (PRG) actions and define CCM1-mediated signaling within the CmPn network. Our findings underscore the vital roles of the CCM1 protein in CmPn/CmP signaling pathways, providing insights into potential transcriptional regulatory mechanisms involved. These results highlight the therapeutic potential of targeting CCM1 for conditions related to the CmPn/CmP networks and significantly enhance our understanding of the molecular mechanisms underlying CmPn/CmP signaling, especially the role of CCM1 within these networks.

## 2. Materials and Methods

### 2.1. Cell Culture, Treatment, Sample Preparations, and Data Acquisition

#### 2.1.1. Cell Culture and Treatment

All three MEF cell lines (CCM1-WT, CCM1-KO, CCM1-KI/96) were cultured at 37 °C and 5% CO_2_ in Dulbecco’s modified Eagle’s medium (DMEM) supplemented with 10% fetal bovine serum (FBS), 2 mM glutamine, and 100 U/mL penicillin/streptomycin as described [28]. We have previously defined progesterone (PRG) action as combine progesterone and mifepristone (PRG+MIF), briefly, when MEF cells reached 80% confluence, cells were treated with either vehicle control (ethanol/DMSO, VEH), or progesterone (PRG) actions (PRG+MIF; 20 µM each), or media only (untreated) for progesterone (PRG) action comparative treatments, as previously described [25,26].

#### 2.1.2. Cell Collection, Protein Sample Preparations, and Data Generation

After the cells from various treatment groups were rinsed and harvested, proteins were extracted from cell lysis using a lysis buffer. Following the extraction of proteins, a purification process utilizing well-established methods was performed to ensure the acquisition of high-quality proteins for subsequent protein data acquisition [24,25,26,29]. To obtain omic data, we employed established protocols [24] and utilized liquid chromatography–tandem mass spectrometry (LC-MS/MS) for peptide analysis. Subsequently, either MaxQuant (V2.6.3.0) or Skyline (V24.1) was employed for data processing in the next phase.

#### 2.1.3. Proteomic Data Acquisition and Processing

Using the proteomics data we obtained, we conducted an analysis of the protein expression patterns in Mouse Embryonic Fibroblasts (MEFs) with three different genotypes of CCM1: Ccm1-WT, Ccm1-KO, and Ccm1-KI. Additionally, we examined how these protein expression patterns responded to combined progesterone treatments. The protein expression patterns of MEFs with three genotypes of CCM1 (Ccm1-WT, Ccm1-KO, and Ccm1-KI) and their response to progesterone (PRG) treatments were analyzed through the use of our acquired proteomics data. For the proteomic data analysis, Sequest was employed to search the UniProt reference protein database, with specific parameters defined, including a fragment ion mass tolerance of 0.020 Da and a parent ion tolerance of 10.0 PPM. In Sequest, we applied a fixed modification of carbamidomethyl on cysteine residues, along with variable modifications of oxidation on methionine residues and acetyl on the N-terminus. To ensure the reliability of peptide and protein identifications based on MS/MS data, Scaffold was utilized. Only identifications surpassing a probability threshold of 95.0% (for peptides) and 99.0% (for proteins), as determined by the Peptide Prophet and Protein Prophet algorithms respectively, were considered valid. The false discovery rate was determined by utilizing the decoy database. In cases where MS/MS analysis alone could not distinguish between proteins, they were grouped together following the principles of parsimony. MaxQuant software (V2.6.3.0) was utilized for protein quantification, and protein abundance was expressed as intensity-based absolute quantification (iBAQ) values. To ensure data quality, low-quality identifications and contaminants were filtered out from the resulting peptide and protein identifications. Differential protein expression analysis was carried out using DEP, Percolator, or ProteinQuant tools to identify proteins that were differentially expressed (DEPs) between the various CCM1 genotypes and their respective responses to the different conditions.

#### 2.1.4. RNAseq Data Acquisition

All RNA samples were obtained from cells using TRIzol (Cat. no. 15596018; Ambion, Waltham, MA, USA). The quantity and purity of the isolated RNA samples were assessed using a Nanodrop spectrophotometer (Nanodrop 2000, Thermo Scientific, Waltham, MA, USA) followed by an Agilent 2100 Bioanalyzer (Agilent Technologies, Santa Clara, CA, USA). The RNA-seq data were generated using the Illumina HiSeq 2000 platform, followed by the processing of the raw RNA-seq data. The data processing workflow included the removal of rRNA reads, filtration of low-quality reads (those with more than 20% of base qualities below 10), and elimination of reads containing adapter sequences and unknown bases (more than 5% N bases). The total clean reads for all samples exceeded 99.5%, with 60–80% of reads successfully mapped to reference genomes. All samples yielded a total of clean reads surpassing 99.5%, and the mapping of 60–80% of these reads to reference genomes was accomplished using the BGI America pipeline based in San Jose, CA, USA. Subsequently, all clean reads were assembled into unigenes, functionally annotated, and evaluated for expression levels and SNPs for each sample. Differential expressed genes (DEGs) were identified, and clustering analysis as well as functional annotations were conducted to provide additional insights. The detailed acquisition of RNAseq data was carried out as previously described [24]. The objective was to uncover the similarities and differences among two genotypes (Ccm1-KO, and Ccm1-KI/96) under either mPR-specific treatments or vehicle control. The consistency in differential expression was analyzed using a Python (v3.12.5) comparison script. For the data analysis, the first step was to remove reads that were mapped to ribosomal RNAs to obtain the raw data. Next, we filtered low-quality reads (reads with more than 20% of bases with a quality score lower than 10), reads with adapter sequences, and reads with unknown bases (more than 5% N bases) to obtain clean reads. We then assembled the clean reads into Unigenes and annotated their functions. The expression levels and SNPs of each sample were then calculated. Finally, we identified DEGs (differentially expressed genes) among the samples and performed clustering analysis and functional annotations.

### 2.2. Omics, Bioinformatics Analysis, and Systems Biology

#### 2.2.1. Pathway Enrichment Analysis

To investigate the gene–gene interactions within the pathways related to the function and signaling of the CCM1 gene, we performed pathway enrichment analysis. During this analysis, proteins were grouped based on significant peptide evidence and annotated using Gene Ontology (GO) terms [30] and the Kyoto Encyclopedia of Genes and Genomes (KEGG) [31]. GO and KEGG pathway analyses are performed to identify enriched pathways associated with differentially expressed genes. The pathway analysis data are integrated using methods like Protein–Protein Interaction Network (PPI) analysis, gene set over-representation analysis (ORA), and comparative omics to identify relationships between proteins and aggregates of signaling pathways based on set intersections across multiple sets [32,33,34].

#### 2.2.2. Systems Biology Analysis

The proteins are grouped based on significant peptide evidence and annotated with Gene Ontology (GO) terms [30] and the Kyoto Encyclopedia of Genes and Genomes (KEGG) database [31]. GO and KEGG pathway analyses are performed to identify enriched pathways associated with differentially expressed genes. The pathway analysis data are integrated using methods like Protein–Protein Interaction Network (PPI) analysis, gene set over-representation analysis (ORA), and comparative omics to identify relationships between proteins and aggregates of signaling pathways based on set intersections across multiple sets [32,33,34].

This study conducted basal-level statistical analysis to determine the differential expressions of various components. *T*-tests were performed on differentially expressed proteins in multiple clusters from various sample comparisons. This study included three pairs for varying levels of CCM1 and three pairs for the effect of different genetic backgrounds under PRG actions. Each experimental set was biologically replicated three times. The *t*-tests were conducted based on quantitative values without multiple test correction, using a significance level of *p* < 0.05, with fold change by category, and followed by normalization. The results were compiled into an Excel sheet, which also included basal-level statistics for each pathway component.

#### 2.2.3. ML-Aided Transcriptional Factors (TF) Prediction Analysis

To enhance the efficiency of transcription factor prediction, we developed our approach to combine the utilization of Evolutionary Scale Modeling (ESM) with a cost-sensitive Support Vector Machine (SVM), with the following three key steps: (1) Evolutionary Scale Modeling (ESM). ESM is an advanced transformer-based language model designed specifically for proteins. It has undergone training on an extensive corpus containing 250 million protein sequences [35]. The main objective of this training was to equip the model with the ability to directly predict protein structure, function, and various properties from individual sequences using masked language modeling. Due to its inherent flexibility, ESM is exceptionally well-suited for fine-tuning a wide range of tasks that necessitate protein sequences as input. At the heart of our methodology lies the utilization of ESM to generate highly representative encodings of protein sequences, which are subsequently employed as input for our prediction model. In this study, we specifically utilized employ ESM-2, a cutting-edge protein language model that has demonstrated superior performance compared to all tested single-sequence protein language models in various structure prediction tasks. Moreover, ESM-2 enables the prediction of atomic resolution structures [36]. Its capabilities in comprehending and predicting protein interactions and functions are unparalleled. (2) The cost-sensitive SVM. SVM is a classifier known for its robustness in distinguishing classes within the feature space by identifying an optimal hyperplane. However, traditional SVMs can face challenges when dealing with imbalanced datasets, as they do not take into account the varying costs associated with misclassifying examples from minority classes. To tackle this issue, we adopt a cost-sensitive SVM approach, where the misclassification costs for each class are assigned based on their inverse proportion to their frequencies. This approach ensures a more equitable treatment of minority classes and addresses the limitations of traditional SVMs [37]. (3) Experimental Setup and Data Preprocessing. Our ML-assisted experimental setup involves conducting experiments using a 10-fold cross-validation strategy on a dataset comprising 3738 protein sequences. The dataset consists of 2784 samples classified as Non-Transcription Factors (NTFs) and 954 samples classified as Transcription Factors (TFs). To ensure consistency across sequences, we apply truncation or padding techniques to achieve a standardized length of 1000 amino acid residues. This preprocessing step ensures that all sequences are of the same length for further analysis and model training.

## 3. Results

### 3.1. Differential Expressed Proteins between Mouse Embryonic Fibroblasts (MEFs) with Different CCM1 Genotypes

The experiment analyzed differentially expressed protein profiles in MEFs with three distinct CCM1 genotypes, classified as CCM1-WT, CCM1-KO, and CCM1-KI/96. These genotypes exhibited significantly different expression levels of CCM1 protein. These genetic variations displayed notably distinct levels of CCM1 protein expression, as previously documented [4,5,6,28,38] and verified by our laboratory prior to this experiment.

#### 3.1.1. Differentially Expressed Protein (DEP) Profiles in the MEFs with Different CCM1 Genotypes

In this comparative proteomic analysis, a Venn diagram was employed to identify differentially expressed proteins (DEPs) in three pairs of CCM1 genotypes: wild-type MEFs with low endogenous CCM1 expression (CCM1-WT) versus ectopically overexpressed CCM1 (CCM1-96), CCM1-depletion genotype (CCM1-KO) versus CCM1-WT genotype, and CCM1-overexpressed (CCM1-96) genotype versus CCM1-WT genotype. The analysis revealed that 559 DEPs were identified in the first genotype pair, with 205 proteins (19.4%) unique to this comparison (Figure 1A, blue circle). Additionally, 487 DEPs were identified in the second genotype pair, with 120 DEPs (11.3%) specific to this comparison (Figure 1A, yellow circle). Furthermore, 621 DEPs were defined in the third genotype pair, with 194 DEPs (18.3%) exclusive to this comparison (Figure 1A, pink circle). Out of the total 1059 identified proteins, 68 (6.4%) were shared across all three genotype pairs, 113 (10.7%) were shared between the first (blue) and second (yellow) genotype pairs, 173 (16.3%) between the first (blue) and third (pink) pairs, and 186 (17.6%) between the second (yellow) and third (pink) pairs (Figure 1A). The DEPs were further categorized as upregulated (red-colored bar) and downregulated (blue-colored bar) in each comparative CCM1 genotype pair. The results indicated more upregulated genes were exclusively identified in the first genotype pair, while more downregulated genes were observed in the second and third genotype pairs (Figure 1B). A heatmap displayed significant differences in DEPs among MEFs with the three different CCM1 genotype pairs, highlighting their distinct expression profiles of proteins (DEPs) (Figure 1C). Similar trends in DEPs were also depicted in volcano plots (Appendix A).

#### 3.1.2. Differential Signal Pathways in the MEFs with Different CCM1 Expression Levels

To uncover the biological significance behind the differentially expressed proteins (DEPs), pathway enrichment analyses were conducted to group genes and identify affected pathways. However, due to the limited number of identified peptides and incomplete pathway coverage, protein data may not fully capture pathway activity, especially when compared to the more sensitive genomic approaches used in other high-throughput technologies. Consequently, there is a significant dataset-dependent impact on the effectiveness of various pathway enrichment methods [39]. Therefore, it is strongly recommended to adopt an integrative approach that involves multiple pathway enrichment analyses using different databases [40]. GO and KEGG pathway enrichment analyses were performed in this project. GO pathway enrichment data were further analyzed using proteomic Gene set overrepresentation analysis (ORA) plots. Comparative ORA plots illustrated enriched pathways among MEFs with different CCM1 protein expression levels (Figure 1D). Similarly, a KEGG ORA plot provided a comparative view of enriched KEGG pathways among MEFs with varying CCM1 expression (Figure 1E). The results revealed that protein folding, processing, and degradation pathways were the main enriched pathways influenced by cellular CCM1 protein levels. This finding was consistent in both GO and KEGG pathway analyses (red-framed pathways, Figure 1D,E). Another enriched pathway involved amino acid, carbohydrate, and lipid metabolism (blue-framed pathways, Figure 1E). GO and KEGG pathway analyses provided valuable insights into functional variations in MEFs with different CCM1 expression levels, revealing underlying mechanisms and consequences of these changes (Figure 1D). Notably, the analysis indicated a higher proportion of downregulated genes in the identified pathways.

### 3.2. Differential Expressed Proteins between Mouse Embryonic Fibroblasts with Different CCM1 Genotypes under Progesterone (PRG) Actions

This experiment aimed to analyze differentially expressed protein profiles in MEFs with three CCM1 genotypes in response to mPR-specific PRG actions by comparing the PRG-treated and control groups.

#### 3.2.1. Differentially Expressed Protein (DEP) Profiles in the MEFs with Different CCM1 Genotypes in Response to PRG-Specific Actions

The Venn diagram analysis showed that 661 differentially expressed proteins (DEPs) were identified in the comparative pair between progesterone (PRG) treated and vehicle control groups among three different CCM1-genotypes: MEFs with endogenously low expression (CCM1-WT), total depletion (CCM1-KO), and ectopically excessive expression (CCM1-KI/96) of CCM1. Among these, 212 proteins (20%) were unique to the CCM1-KO genotype (Figure 2A, yellow circle). Additionally, 218 DEPs (20.6%) were identified in the CCM1-WT genotype (Figure 2A, blue circle), with 115 DEPs (10.9%) being specific to the CCM1-KI/96 genotype (Figure 2A, pink circle). Out of the 1059 total identified proteins, 12 (1.1%) were shared by all three genotypes, 48 (4.5%) were shared by the CCM1-WT and KO genotypes, 29 (2.7%) were shared by the CCM1-KO and KI/96 genotypes, and 27 (2.5%) were shared by the CCM1-WT and KI/96 genotypes (Figure 1A). The DEPs were further classified as upregulated (red-color bar) and downregulated (blue-color bar) in each CCM1 genotype under progesterone (PRG) actions. The results showed that more upregulated genes were exclusively identified in the CCM1-KO genotype, while more downregulated genes were observed in the CCM1-WT and CCM1-KI/96 genotypes (Figure 2B). A heatmap showcased the significant differences in DEPs among MEFs with three distinct CCM1 genotypes under progesterone (PRG) treatment, highlighting their markedly differential protein expression profiles (Figure 2C). Similar trends in DEPs were also depicted in volcano plots (Appendix A).

Similarly, the RNAseq experiment identified differentially expressed genes (DEGs) in MEFs with two contrasting CCM1 genotypes: a CCM1-total depletion genotype (CCM1-KO) and an ectopically overexpressed CCM1 genotype (CCM1-KI/96). These genotypes showed a distinctly binary ON/OFF expression of the CCM1 protein. We explored how these opposite CCM1 expression patterns affected the DEG profiles in MEFs in response to membrane-specific (mPR-specific) progesterone (PRG) actions.

In this comparative genomic analysis, a Venn diagram was first used to identify differentially expressed genes (DEGs) in two opposite CCM1 genotypes, classified as CCM1-total depletion genotype (CCM1-KO), and ectopically overexpressed CCM1 genotype (CCM1-KI/96). The analysis using Venn diagrams showed that when treated with membrane-specific (mPR-specific) progesterone (PRG) actions, a total of 15,707 genes exhibited differential expression in the CCM1-KO genotype pair (Figure 2D). Conversely, in the CCM1-KI/96 genotype with ectopically excessive expression of CCM1, 15,486 genes showed differential expression (Figure 2E). The HeatMap plot provided a visual representation of the variation in gene expression between MEFs with two different CCM1 genotypes under progesterone (PRG) treatment, indicating significant differences in gene expression profiles (Figure 2F). Similar trends in DEPs were also depicted in volcano plots (Appendix A). These findings suggest that the presence of the CCM1 gene may enhance the induction of differentially expressed genes (DEGs) through the specific action of mPR-related PRG. These findings indicate a strong correlation between CCM1 and PRG-mediated activities through a mPR-specific mechanism within the CmPn signaling network, operating through unique pathways at the transcriptional level.

#### 3.2.2. Different Pathway Response in the MEFs with Three Different CCM1 Genotypes under Progesterone (PRG) Actions

To investigate the effects of CCM1 on the responsive DEP profiles in MEFs under progesterone (PRG) treatment, we need to identify the distinct pathways by comparing three MEFs with different CCM1 expression patterns: depleted (CCM1-KO), endogenously low (CCM1-WT), and ectopically overexpressed CCM1 (CCM1-96) under PRG treatment, in comparison to vehicle controls. To achieve this, we utilized GO pathway enrichment data and performed proteomic gene set over-representation analysis (ORA) (Appendix A). First, we conducted comparative GO ORA to assess differentially expressed proteins (DEPs) in PRG-treated MEFs compared to their corresponding vehicle controls, considering three genotypes of CCM1 (KO, WT, and overexpression-9/6 MEF lines), which represent various levels of CCM1 protein expression (Figure 2G and Appendix A). This analysis was repeated with KEGG ORA under the same conditions (Figure 2G and Appendix A). Additionally, we performed comparative GO/KEGG ORA to assess differentially expressed RNAs (DEGs) in PRG-treated MEFs compared to their corresponding vehicle controls, considering two different CCM1 genotypes (KO, 9/6 MEF lines) of CCM1 protein expression (Figure 2H and Appendix A).

Our results indicated that by removing the originally enriched protein folding, processing, and degradation pathways (red framed pathways in Figure 1D,E) and the common amino acid, carbohydrate, and lipid metabolism pathways (blue framed pathways in Figure 1E), significant enriched transcriptional and translational signal pathways were identified in response to PRG actions. These findings suggest that the underlying mechanisms of the CCM1 protein in the CmPn signaling network in response to PRG actions involve both transcriptional and translational signal pathways. These systems’ biological findings indicate that maintaining an intricate balance of CCM1 protein levels is crucial for its involvement in the CmPn signaling network in response to progesterone (PRG) actions. The underlying mechanisms encompass both transcriptional and translational signal pathways. Disruptions, caused by either too little (CCM1 null) or excessive amounts (CCM1-9/6, overexpression of CCM1), can disturb this delicate equilibrium, leading to disruptions in the CmPn signaling pathway.

### 3.3. Pathways Related to Non-mPR PRG Actions Modulated through CCM1 Proteins within CmPn Network Can Be Eliminated by Using Established Proteomic Data

The glucocorticoid receptor (GR) and nuclear progesterone receptor (nPR) share the same DNA sequence in their promoter regions [41], and also have a common antagonist, mifepristone (MIF or RU486), as their ligand [42,43]. This has led to the application of MIF in various clinical therapies [42,43,44,45,46,47]. Numerous cell types have been found to express both the glucocorticoid receptor (GR) and nuclear progesterone receptor (nPR) [41,48,49,50,51,52,53]. The combined actions of the classic and non-classic PRG receptors (nPR/mPR) have been studied through the use of combined treatment with PRG and MIF [23,25,26,53,54,55,56]. There exists only one source of well-established proteomic data for U2OS cells treated with Mifepristone (MIF, RU 486) [57]. U2OS cells, a human sarcoma with an epithelial origin that expresses both GR and nPR [58,59], can undergo EMT to link with fibroblasts [60]. MEFs, fibroblasts that also express both GR and nPR, are similarly affected by Mifepristone [57,61,62,63]. This is because MIF acts as an antagonist for both GR and nPR [57,58,59,60,61,62,63]. In this section, we will use this dataset as a filter to remove any “antagonist” effects resulting from non-mPR-specific PRG actions, such as PRG actions through GRs and nPRs (Appendix A).

Through the Venn diagram analysis, we observed that under progesterone (PRG) treatments, across MEFs with three different CCM1 genotypes (CCM1-WT with low expression, CCM1-KO with total depletion, and CCM1-KI/96 with excessive expression), a total of 876 differentially expressed proteins (DEPs) and 6424 differentially expressed RNAs (DEGs) were identified after filtration (Appendix A). Out of the 876 filter-selected (mPR-specific) DEPs, 224 (25.6%) were unique to the CCM1-KO genotype (Appendix A, yellow circle), 262 (29.9%) were identified in the CCM1-WT genotype (Appendix A, blue circle), and 151 (17.2%) were specific to the CCM1-KI/96 genotype (Appendix A, pink circle). Additionally, 52 (5.9%) were common to all three genotypes, 79 (9.0%) were shared by the CCM1-WT and KO genotypes, 42 (4.8%) were shared by the CCM1-KO and KI/96 genotypes, and 66 (7.5%) were shared by the CCM1-WT and KI/96 genotypes (Appendix A). Similarly, among these mPR-specific DEGs, 1230 (19.1%) were unique to the CCM1-KO genotype (Appendix A, red circle), and 3242 (50.5%) were specific to the CCM1-KI/96 genotype (Appendix A, blue circle), while 1952 (30.4%) were shared by the CCM1-KO and KI/96 genotypes. The volcano plots displayed the upregulated (red dots on the right) and downregulated (blue dots on the left) DEPs in MEFs with the CCM1-KO genotype (left panel), CCM1-WT genotype (middle panel), and CCM1-KI/96 genotype under progesterone treatment (PRG) (Appendix A). Similar volcano plots displayed the upregulated (red dots on the right) and downregulated (blue dots on the left) DEPs in MEFs with the CCM1-KO genotype (left panel), and CCM1-KI/96 genotype in response to PRG action. Additionally, the heatmap illustrated a similar pattern of significant differences in DEPs (Appendix A) and DEGs (Appendix A) among MEFs with different CCM1 genotypes under PRG actions, emphasizing the distinct expression profiles at both transcriptional and translational levels after filter selection.

A comparative ORA plot integrating both GO and KEGG analyses illustrated the enriched pathways as the filter. The identified non-mPR-specific pathways from DEPs modulated by the CCM1 protein include DNA replication and repair pathways (such as single-stranded DNA helicase activity, helicase activity, DNA helicase activity, and catalytic activity acting on DNA), as well as cell–cell adherent junctions (cadherin binding), which are prominently associated with cell proliferation (Figure 3A and Appendix A). Similarly, a comparable approach was applied to RNAseq data to generate the nPR response filter. This filter identified GO/KEGG enriched pathways, including angiogenesis (highlighted in black), tumorigenesis (highlighted in red), cell proliferation and performance (highlighted in green), RNA processing (highlighted in light blue), and inflammatory response (highlighted in purple) (Figure 3B and Appendix A). These findings serve as the non-mPR-specific PRG actions filter for RNAseq data. The results indicate that most of the identified pathways associated with PRG actions and regulated by CCM1 (Figure 2G,H) fall into the second filter, suggesting their predominant involvement in non-mPR-specific PRG actions within MEF cells. Our findings indicate that the majority of PRG action-response DEPs and DEGs mediated by CCM1 within the CmPn network are associated with non-mPR-specific receptors, such as the glucocorticoid receptor (GR) and nuclear progesterone receptor (nPR) (Figure 3), which will be filtered out.

### 3.4. Identification of mPR-Specific PRG Pathways Modulated through CCM1 Proteins in the CmPn Signal Network

In previous experiments, we established two sets of filters to identify mPR-specific PRG pathways through a two-step filtration process. The first step filtered pathways based on CCM1 protein expression levels, while the second step filtered out non-mPR-specific PRG actions. Our findings show that mPR-specific PRG actions mediated by CCM1 within the CmPn network are less common than PRG actions through non-mPR-specific receptors, such as the glucocorticoid receptor (GR) and nuclear progesterone receptor (nPR).

A Venn diagram analysis under mPR-specific PRG actions revealed 18 differentially expressed proteins (DEPs) and 103 differentially expressed RNAs (DEGs) across MEFs with different CCM1 genotypes (CCM1-WT, CCM1-KO, and CCM1-KI/96) after filtration (Appendix A). Among the 18 mPR-specific DEPs, 2 (11.1%) were unique to CCM1-KO, 7 (38.9%) to CCM1-WT, and 2 (11.1%) to CCM1-KI/96. Additionally, 3 (16.7%) were shared by CCM1-WT and KO, 3 (16.7%) by CCM1-KO and KI/96, and 1 (5.6%) by CCM1-WT and KI/96 (Appendix A). Among the mPR-specific DEGs, 20 (19.4%) were unique to CCM1-KO, 57 (55.3%) to CCM1-KI/96, and 26 (25.2%) were shared by CCM1-KO and KI/96 (Appendix A). Heatmaps illustrated similar changes in DEPs (Appendix A) and DEGs (Appendix A) among different CCM1 genotypes, emphasizing distinct expression profiles for mPR-specific PRG actions at both transcriptional and translational levels.

#### 3.4.1. Proteomic Identification of mPR-Specific PRG Pathways Regulated by CCM1 Proteins

mPR-specific PRG actions show unique sensitivity to CCM1 regulation. Proteomic data analysis of both GO and KEGG pathways revealed that PRG pathways specific to mPR are enriched with molecular chaperones, particularly HSPs-70 and 90, which interact with steroid receptors in the absence of ligands. Other identified pathways include axonal guidance transmembrane signaling receptors (e.g., semaphorin receptor), cell–cell junctions and communications (e.g., gap junction), and the membrane “chaperone” phospholipid (phosphatidylethanolamine, PE), known to contribute to multiple signaling pathways like MAP kinase (MAPK) (Figure 4A and Appendix A). Notably, certain pathways such as cellular autophagy (TORC2 complex) and translational activity (ribosome) were excluded during the filtering process (highlighted in yellow), as indicated by the filters (Figure 1D,E and Figure 3A,B).

#### 3.4.2. Transcriptional Profiling of mPR-Specific PRG Pathways Regulated by CCM1 Proteins

Similarly, GO and KEGG pathway analysis using RNAseq data revealed the enrichment of multiple pathways specific to mPR. After excluding false-positive pathways, such as RNA process protein folding (highlighted in yellow), a prominent pathway that emerged was the calcium channel pathway (highlighted in green). Additional pathways identified included cellular performance (highlighted in light blue) and protein synthesis (highlighted in red) (Figure 4B and Appendix A).

#### 3.4.3. Omics Analysis of mPR-Specific PRG Pathways by CCM1 Proteins

In our combined analysis, we conducted ORA using GO and KEGG pathway analysis, integrating omics data. After eliminating components identified by the two filters (highlighted in yellow), we observed that most pathways align with those previously described (highlighted in green), including the prominent calcium channel pathway, molecular chaperones, axonal guidance pathway, and cellular autophagy. However, we also identified unexpected pathways, such as inflammatory pathways (highlighted in blue) and transcriptional factors (highlighted in red), suggesting the potential involvement of transcriptional regulation in CCM1-mediated mPR-specific PRG actions (Figure 4C and Appendix A).

### 3.5. Discovery of Novel Transcription Factors in mPR-Specific PRG Pathways Regulated by CCM1 in the CmPn Signaling Network

In this study, our aim was to identify cellular signaling changes induced by mPR-specific action associated with CCM1. Building on significant findings from previous experiments regarding potential transcriptional regulation linked to CCM1 protein-mediated mPR-specific progesterone (PRG) action, our next step involved employing machine-learning techniques to predict the role of transcription factors (TFs) in this signaling pathway, using our newly developed ESM-based Cost-Sensitive SVM (ESM-CS-SVM) approach. Employing a 10-fold cross-validation approach, our machine-learning TF prediction model achieved impressive performance metrics, with an average F1-score of 0.9478, specificity of 0.9627, sensitivity of 0.9513, and balanced accuracy of 0.9570. These parameters underscore the effectiveness of our ESM-based, cost-sensitive SVM approach in predicting transcription factors from protein sequences. Our future efforts will concentrate on optimizing the model and enhancing its capacity to handle diverse and complex datasets. Overall, our machine learning techniques achieved a confidence interval exceeding 95% for specificity and sensitivity (Table 1A,B). Using our optimized ESM-CS-SVM model, we identified multiple candidate TFs across approximately 50% of the testing epochs, such as B cell receptor-associated protein 31 (BCAP31) from proteomic data, which align well with the newly identified inflammatory response pathway (Figure 4A–C and Appendix A). In our final analysis of these identified TFs, we conducted ORA using GO and KEGG pathway analysis, integrating omics data. We observed that the majority of pathways are indeed transcriptional regulatory pathways that modulate our previously described pathways (Figure 5, highlighted in red). Additionally, some TFs are involved in controlling other cellular pathways, such as membrane transporters (highlighted in green), translational regulation (highlighted in blue), and energy supply (highlighted in purple) (Figure 5).

## 4. Discussion

Initially, isogenic mouse embryonic fibroblasts (MEFs) were established from E8.5 mouse embryos of WT and Ccm1-knockout (KO) mice. Additionally, Ccm1-knockin (KI) MEFs (CCM1-KI/96) were created by infecting Ccm1-KO MEFs with a lentiviral vector encoding human KRIT1 (CCM1-transduced, Lv-KRIT1) [4]. By utilizing these three MEFs with varying Ccm1 gene dosages (WT, Ccm1-KO, Ccm1-9/6), it was found that CCM1 loss-of-function (LOF) leads to increased susceptibility to oxidative DNA damage, induction of DNA damage sensors and repair genes, and apoptotic response. These findings suggest that CCM1 may play a role in maintaining intracellular reactive oxygen species (ROS) homeostasis to prevent ROS-induced cellular dysfunctions [4]. Moreover, CCM1 prevents upregulation of c-Jun induced by oxidative stimuli [5] and inhibits abnormal activation of the Nrf2 stress defense system and its downstream effectors HO-1 and Glo1 [7]. The MEF toolset was also used to validate the discovery that CCM1 LOF is associated with increased expression of Vegfa and subsequently elevated activation of its receptor, VEGFR2. This leads to loss of barrier function by disrupting the β-catenin-VE-cadherin interaction in vascular endothelial cells (ECs) [8].

The blood–brain barrier (BBB) tightly controls molecular exchanges between the blood and the central nervous system (CNS), making it a crucial interface to understand for addressing neurological conditions, especially hemorrhagic stroke [64,65]. Despite the lack of readily available clinical agents or measures to prevent BBB leakage or repair it, numerous epigenetic mechanisms or regulators that are either protective or disruptive to BBB components have been identified, indicating potential therapeutic opportunities to meet this challenge in the future [66]. Inflammatory events are widely recognized as a major threat to the integrity of the blood–brain barrier (BBB) [12,13]. Steroids are commonly used as therapeutic agents for inflammatory diseases and as treatments for edema [11] in various human conditions, including those related to BBB disorders [11,12,13,14,15]. However, their use remains a subject of ongoing debate [67].

In the absence of hormones, steroid receptors as hormone-inducible transcription factors are bound to chaperones, which was identified as one of the mPR-specific pathways mediated by CCM1 within the CmPn signal network in this study (Figure 4C). According to recent findings, a novel signaling network called the CSC-mPR-PRG-nPR/CSC-mPR-PRG (CmPn/CmP) operates within endothelial cells (ECs), while the CmPn signaling network is present in nPR(+) cells and the CmP signaling network is also present in nPR(−) cells [23,24,25,26,53,54,55,56,68]. Upon binding to a hormone, the receptor undergoes a conformational change and translocates to the nucleus as a transcriptional factor [43]. Many steroid receptors share common ligands and can coordinate their physiological function by competing for the same ligand and DNA motifs as a transcriptional co-factor [51], such as mifepristone, a common antagonist for both glucocorticoid receptor (GR) and nuclear progesterone receptor (nPR).

Previous data of signal transduction modulated by the CCM signal complex (CSC), suggested that perturbed CSC after depletion of one of three CCM (CCM 1, 2, 3) genes, leads to blood vessel cell junction organization disruption [24,26,69,70]. Comparative omics data across multiple models also provided further evidence that the CSC can couple both classic nuclear progesterone receptor (nPR) and non-classic membrane progesterone receptor (mPR) to form a large CmPn/CmP signaling network for progesterone-mediated cellular actions, which may affect cell junction organization and lead to compromised BBB [24,25,26,29,53,55,71,72,73,74]; this may affect cell junction organization and lead to compromised BBB [25]. In this study, we utilized three MEFs with differential expression levels of the CCM1 toolset to perform proteomic analysis to elucidate Ccm1-mediated signaling, specifically through non-classic membrane progesterone receptor (mPR) within the CmPn signaling network.

Transcription factor (TF) prediction plays a vital role in gene expression analysis as it enables us to understand the regulatory mechanisms governing gene expression. TFs are proteins that bind to specific DNA sequences and control the transcription of target genes, thereby influencing gene expression levels. Predicting TFs involves computational methods that utilize sequence information, DNA binding motifs, and other relevant features to identify potential TFs within a gene set or genomic regions. By predicting TFs, we gain insights into regulatory networks, unravel the complexities of gene expression, and comprehend the functional roles of TFs in cellular processes and disease. Through the integration of modern statistical methods and artificial intelligence, specifically deep learning, we have identified BCAP31 as a promising transcription factor. BCAP31, the most abundant protein in the endoplasmic reticulum (ER) [75], is regulated by CCM genes but negatively influenced by PRG-mediated mPR actions. The association of BCAP31 with CCM genes reveals a novel signaling cascade. Further investigations are required to validate this interaction and enhance our understanding of BCAP31’s functionality and its interplay with CCM genes. These contexts encompass endothelial cells participating in vascular angiogenesis, and various types of tumor cells found in breast cancers and liver cancers [24,25,26,29,53,55,76,77].

This experiment explored protein and gene expression variations in mouse embryonic fibroblasts (MEFs) with different CCM1 genotypes and their response to progesterone (PRG). Distinct protein expression profiles were observed among CCM1 depletion (knockout, KO), CCM1-endogenously low (WT), and ectopically excessive expression CCM1 (CCM1-KI/96) MEFs, with some overlapping proteins. These proteins were categorized as upregulated or downregulated in each genotype pair. Pathway analysis using gene ontology (GO) and the Kyoto Encyclopedia of Genes and Genomes (KEGG) revealed the impact of CCM1 protein expression on protein folding, processing, degradation, metabolism, signaling, and transcriptional and translational pathways. This study also investigated gene expression differences between CCM1-KO and CCM1-KI/96 MEFs under progesterone (PRG) influence, highlighting the role of CCM1 in transcriptional and translational signaling pathways. Our findings suggest that CCM1 is involved in the signaling network associated with progesterone (PRG) actions with the CmPn signal network.

In summary, our study adopts an innovative approach by utilizing omics data to establish two layers of pathway filters, effectively eliminating background noise attributed to the glucocorticoid receptor (GR) and nuclear progesterone receptor (nPR). This methodology allows us to elucidate the intricate interactions between CCM1 molecular function and progesterone (PRG) actions within the CmPn/CmP signal network. Through these investigations, we have successfully illustrated the robust connection between CCM1 and progesterone (PRG) actions, particularly those specific to the membrane progesterone receptor (mPR) within the CmPn signal network. These groundbreaking discoveries collectively present a new viewpoint, indicating that directing interventions toward CCM1 could potentially become a promising therapeutic approach for diverse human conditions. This includes vascular angiogenesis related to the disrupted blood–brain barrier (BBB) in humans [25], as well as various types of tumor cells influenced by CmPn signal networks, as described previously [25,26,53,55,77]. Both vascular tumors (hemangiomas) and vascular malformations (including venous malformations (VMs), lymphatic malformations (LMs), and arteriovenous malformations (AVMs), etc.) usually present cervicofacial manifestations that impact both function and aesthetics [78]. The application of therapeutic strategies for hemangiomas, such as beta blockers and steroids, has been highly successful [78]. However, recent attempts to apply the same drug therapies to vascular malformations have produced conflicting results. Our study aims to offer a scientific explanation for the differences between these two cerebral vascular anomalies, with the potential to pave the way for new therapeutic approaches.

Finally, it is important to acknowledge the limitations of this study. Although our findings are novel and intriguing, we must point out that the two filters we employed were derived from different mammalian species and cell lines due to the limited research available in this area. This discrepancy raises valid concerns about the stringency of our two-layer system—specifically, whether it is too strict and may exclude genuine targets, or too lenient and allows excessive background noise. Future research will be necessary to thoroughly address and contest these potential weaknesses.

## 5. Conclusions

The findings of this paper reveal the role of CCM1 in regulating cellular processes under different PRG actions. This study found significant differences in gene and protein expression between different genotypes of CCM1 under PRG actions, indicating that CCM1 plays a critical role in regulating the CmPn network. The enriched pathways determined by cellular CCM1 protein dosage were mainly related to protein folding, processing, and degradation. However, after removing the CCM1 protein dosage-dependent pathways, this study identified significantly enriched transcriptional and translational signal pathways in response to progesterone (PRG) actions. Next, using established filters, this study defined the unique pathways involved in mPR-specific PRG actions mediated by CCM1 protein within the CmPn signal network. Finally, utilizing our own ML/DL-algorism, we generated a list of identified transcription factors (TFs) from our ORA results and elucidated the possible transcriptional regulatory mechanism for unique signal pathways involved in mPR-specific PRG actions mediated by CCM1 protein within the CmPn signal network.

## Figures and Tables

**Figure 1 diagnostics-14-01895-f001:**
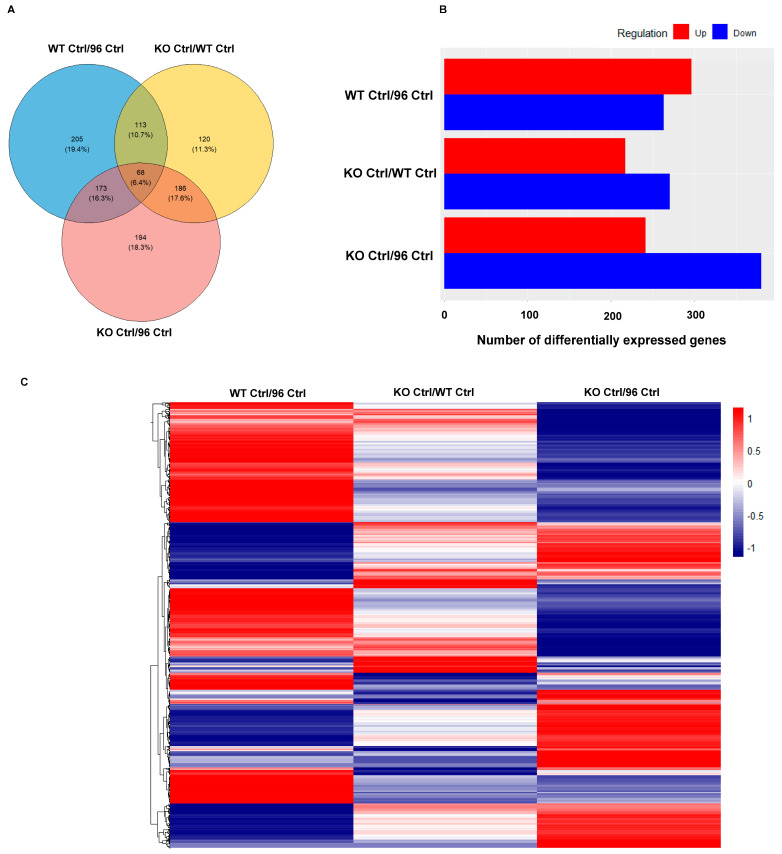
Differentially expressed protein (DEP) profiles in the MEFs with three different genotypes of CCM1, including depletion (knockout, KO), endogenously low (WT), and ectopically excessive expression (knockin, KI/96) of CCM1. The protein profiles of mouse embryonic fibroblasts (MEFs) were examined for differential expression in relation to CCM1 depletion (knockout, KO), endogenous low levels (WT), and excess expression through knockin (KI/96). These DEPs were analyzed to investigate the impact of genetic backgrounds on protein expression profiles in MEFs. (**A**) To compare the differentially expressed protein (DEP) profiles of three pairs of mouse embryonic fibroblasts (MEFs) with total depletion (CCM1KO), endogenous low expression levels (CCM1—WT), or excess expression through knockin (CCM1KI/96) of CCM1, a Venn diagram was utilized. This diagram illustrates the number of DEPs with significant expression changes between the paired comparisons of CCM1 genotypes, as well as the number of specifically expressed proteins between two different CCM1 genotypes. (**B**) To depict the transcriptome profiles of differentially expressed proteins (DEPs) in mouse embryonic fibroblasts (MEFs) across three comparative pairs of CCM1 genotypes, a bar diagram was employed. The diagram showcases the number of upregulated (represented by red bars) and downregulated (represented by blue bars) DEPs between two different CCM1 genotypes. (**C**) A heatmap was used to visualize the differential expression levels and profiles of DEPs in three comparative pairs of CCM1 genotypes in mouse embryonic fibroblasts (MEFs) in response to PRG actions. The heatmap shows the upregulation (red lines) and downregulation (blue lines) of DEPs for each comparative genotype pair across the three different CCM1 genotypes under PRG treatment. The DEPs heatmap was created using *t*-test statistical analysis and visualized with clustering software, as detailed in the Methods section. (**D**) The ORA results highlighting core-enriched differentially expressed proteins (DEPs) in GO pathways were presented for both progesterone (PRG) treatment and vehicle control across three CCM1 genotypes. These genotypes are total depletion of CCM1 (CCM1-KO, left), endogenous low levels of CCM1 (CCM1-WT, middle), and ectopic overexpression of CCM1 through CCM1-knockin (CCM1-KI/96, right). *p*-values were adjusted across all three CCM1 genotypes. (**E**) Similarly, in the comparative ROA plots, red circles represent DEPs with lower significance in adjusted *p*-value, while blue circles indicate DEPs with higher significance in adjusted *p*-value. The size of the circles corresponds to the GeneRatio value, which measures whether genes from predefined sets (e.g., genes in the untreated group within a specific GO term or KEGG pathway) are more present than expected (over-represented) in another subset of the data (e.g., treated).

**Figure 2 diagnostics-14-01895-f002:**
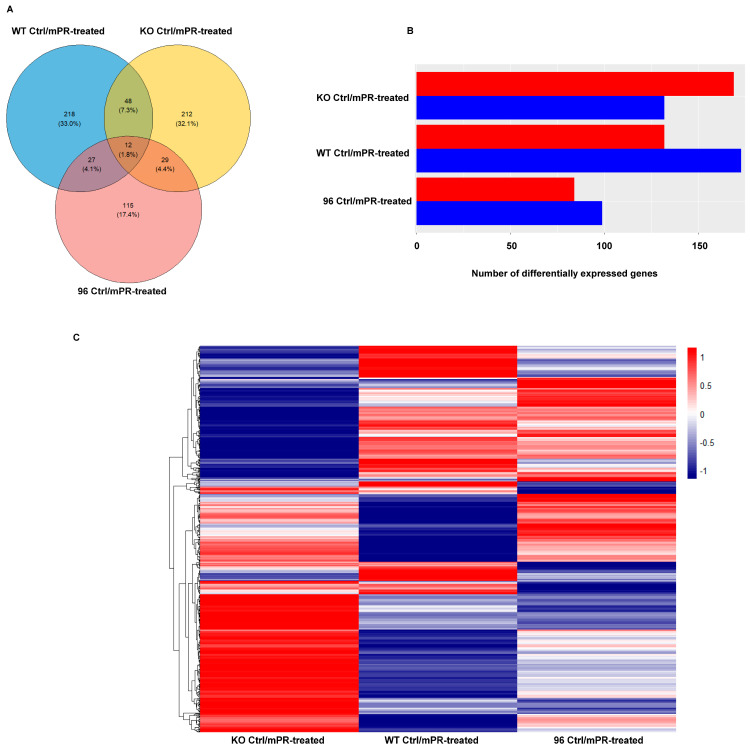
A multi-omics approach to analyzing proteomic and RNAseq data under various levels of CCM1 expression in response to progesterone (PRG) actions. Both protein and RNA expression profiles of MEFs across three different levels of CCM1 expression were investigated. The proteomic data included three distinct genotypes: complete depletion of CCM1 via CCM1-knockout (CCM1-KO), natural low expression levels of CCM1 (CCM1-WT), and ectopic overexpression of CCM1 via CCM1-knockin (CCM1-96). Meanwhile, the RNAseq data covered two levels of expression profiles: complete depletion (knockout) and ectopic overexpression levels of CCM1 via CCM1-knockin (CCM1-96). (**A**) The expression profiles of DEPs in response to PRG actions were investigated in MEFs with three different CCM1 genotypes. A Venn diagram was used to illustrate the specific DEPs affected by progesterone (PRG) treatment compared to vehicle controls across the three CCM1 genotypes in mouse embryonic fibroblasts (MEFs). (**B**) A bar diagram was used to depict the profiles of DEPs in response to PRG actions across three CCM1 genotypes. The diagram shows the distribution of upregulated DEPs (represented by a red bar) and downregulated DEPs (represented by a blue bar). (**C**) A heatmap was employed to visualize the profiles of DEPs in response to PRG actions across three CCM1 genotypes. The heatmap illustrates the distribution of upregulated DEPs (indicated by red lines) and downregulated DEPs (indicated by blue lines). Comparative analysis of DEPs was performed using pairwise *t*-test statistical analysis. (**D**) Similarly, for RNA expression profiling, Venn diagrams were used to compare expression patterns between untreated MEF KO controls and MEF-KO treated with progesterone (PRG) in the CCM1-depletion genotype (MEF KO). The analysis identified 751 unique differentially expressed genes (DEGs) upon exposure to PRG. (**E**) Likewise, for the ectopic overexpression of the CCM1 (MEF 96) genotype, we compared expression patterns between untreated MEF 96 controls and MEF 96 treated with progesterone (PRG). This analysis revealed 573 differentially expressed genes (DEGs) upon exposure to PRG. (**F**) A heatmap was generated to depict the profiles of differentially expressed genes (DEGs) in response to PRG actions across two CCM1 genotypes. The comprehensive analysis involved examining 1900 shared RNAseq profiles between the two CCM1 genotypes to investigate how exposure to PRG influences the regulation of common sequences across different levels of CCM1, both with and without PRG treatment. (**G**) The ORA results, highlighting core-enriched DEPs in both GO and KEGG pathways, were presented to show the differential responses to PRG actions across three CCM1 genotypes (CCM1-KO, left; CCM1-WT, middle; and CCM1-KI/96, right). The analysis plots depict data specifically filtered with CCM1-associated pathways from enriched DEPs in both GO and KEGG approaches (Figure 1D,E), with the original dataset provided in the Appendix A. It is important to highlight that all samples underwent triplicate analysis, and statistical significance was evaluated using a *t*-test, with *p*-values less than 0.05 considered significant. (**H**) Additionally, the RNAseq data underwent ORA similar to the previous approach used for proteomic data. Importantly, we emphasized shared pathways between the RNAseq profiles and the proteomic data by highlighting them with bold borders, specifically focusing on Diabetic cardiomyopathy and Aminoacyl-tRNA biosynthesis. In the comparative ROA plots, the depiction and scale of circles remain consistent with the previous ones and uniform throughout this manuscript.

**Figure 3 diagnostics-14-01895-f003:**
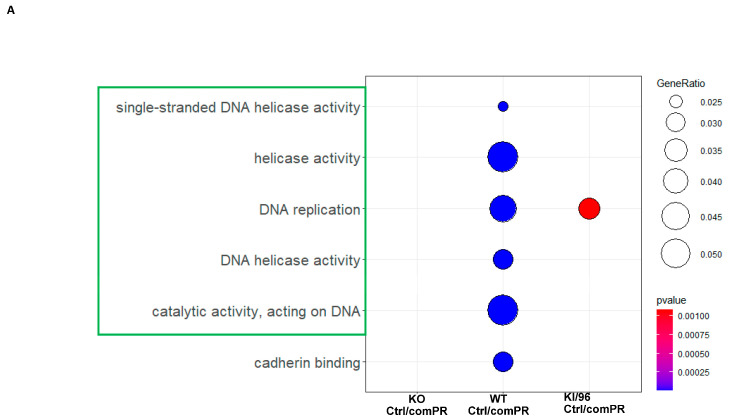
**Generation of Combined Non-mPR PRG Action Filters Using a Multi-Omics Approach.** This experiment illustrates our approach to create a secondary filter associated with non-mPR-specific PRG actions. This filter aimed to identify all pass-through pathways induced by all PRG actions. Given the available data, our initially identified targets were all DEPs and DEGs in response to PRG actions through CCM1, some of which are not mediated through mPR-specific PRG actions. In this experiment, we defined all mPR-specific DEPs and DEGs by employing the filter extracted from an established dataset of DEPs/DEGs in response to mifepristone (MIF, as an antagonist to nPRs/GRs, but agonist to mPRs) treatment, to identify pathways shared by the glucocorticoid receptor (GR) and the classic nuclear progesterone receptor (nPR). It is crucial to reemphasize that MIF functions only as an antagonist to the pathways shared by the GRs and nPRs. However, it acts solely as an agonist and works synergistically with PRG on mPR-mediated signaling (Appendix A), forming the foundation for this experiment. (**A**) Combined GO/KEGG ORA enrichment of DEPs from proteomic data in response to PRG actions through CCM1. In the panel, pathways framed in yellow represent cell proliferation processes, such as DNA replication and DNA repair pathways, including single-stranded DNA helicase activity, and catalytic activity acting on DNA. In contrast, pathways framed in green indicate cell–cell adherent junctions. (**B**) GO/KEGG ORA pathway enrichment of DEGs from RNAseq transcriptional expression data in response to PRG modulation via CCM1. In the panel, pathways highlighted in yellow represent cell proliferation processes, while green frames indicate cell–cell adherent junctions. Additionally, red frames highlight cellular signal transduction factors, purple frames denote inflammatory factors, and black frames indicate angiogenic factors. Statistical significance was assessed using a Student’s *t*-test, with a significance threshold set at *p* < 0.05.

**Figure 4 diagnostics-14-01895-f004:**
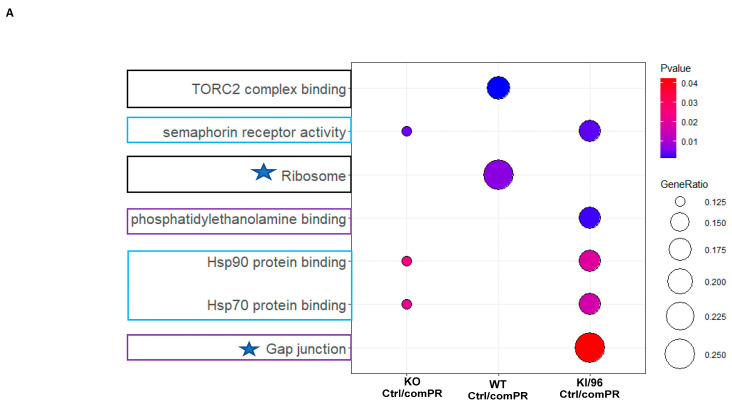
**Identifying mPR-Specific pathways Using combined Non-mPR Progesterone Action Filters.** The objective of this experiment was to identify signaling pathways specifically associated with the membrane progesterone receptor (mPR) by excluding the PRG actions that are not mPR-specific across the distinct CCM1 genotypes defined in the previous step (Appendix A). This approach aimed to generate mPR-specific signaling pathways in response to PRG actions via CCM1, as represented below. (**A**) To visualize the ORA results of core-enriched DEPs from GO and KEGG pathway enrichments, an integrative dot plot was used. The plot displayed data from three CCM1 genotypes that passed the filter for non-mPR-specific PRG actions. Triplicate analysis was conducted, and statistical significance was determined using a Student’s *t*-test with a *p*-value cut-off of less than 0.05. The integrative dot plot highlighted the enriched pathways associated with mPR-specific PRG action, indicated by red-framed, blue-framed, and black pathways. (**B**) Similarly, an integrative dot plot was employed to visualize the ORA signaling pathways for mPR-specific PRG actions of enriched RNAseq data from GO and KEGG pathway enrichments. These data, which underwent triplicate analysis, passed through the filter for non-mPR-specific PRG actions among the two CCM1 genotypes used in the RNAseq portion of this study. The statistical significance of the results was determined using a Student’s *t*-test, with a *p*-value cut-off of less than 0.05. (**C**) Finally, a summarized dot plot was created to visualize the combined ORA signaling pathways for mPR-specific PRG actions, highlighting the core-enriched DEPs and DEGs from both proteomic and RNAseq data.

**Figure 5 diagnostics-14-01895-f005:**
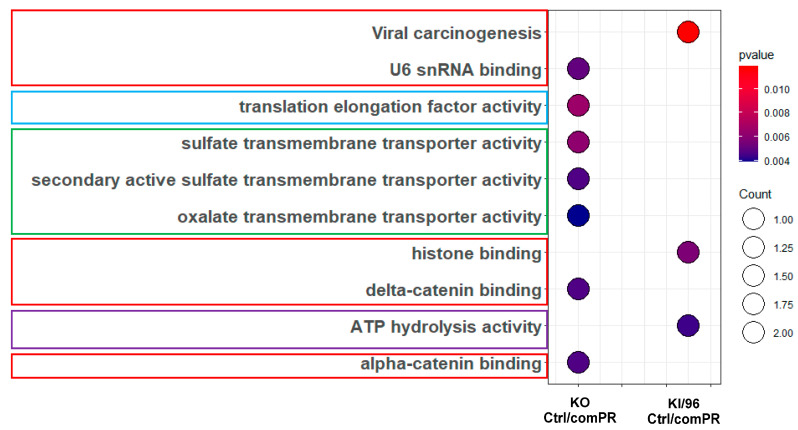
Identifying transcription factors (TFs) in mPR-specific regulatory pathways using ML/DL techniques with enrichment of mPR-specific DEPs and DEGs. After machine learning/deep learning (ML/DL)-based prediction, we identified 12 potential TFs and explored their functional roles using Entrez ID identifiers.

**Table 1 diagnostics-14-01895-t001:** **ML/DL Models Predict Functionality of mPR-Specific action-associated cellular factors**. (**A**) The Evolutionary Scale Modeling and Support Vector Machine (ESM-SVM) models exhibited the highest accuracy among those employed in this study. These models were trained on a dataset comprising 4331 unique FASTA sequences, selected based on achieving an average accuracy of approximately 95% across all tests. (**B**) Furthermore, Convolutional Neural Networks (CNNs) were assessed to ascertain whether they could achieve a comparable 95% accuracy. However, these attempts were unsuccessful. Nevertheless, we compared the results of both models to identify sequences most likely associated with suggested transcription factor functionality.

	A. ESM-SVM Models	B. CNNs Models
**F1 score**	0.948	0.934
**Specificity**	0.963	0.954
**Sensitivity**	0.958	0.947
**Accuracy**	0.961	0.950

## Data Availability

Readers can access the data supporting the conclusions of this study through Appendix A and some omics data are in the process of being deposited into the NIH genomic or proteomic databases repertoire, and can be acquired by contacting the corresponding author.

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
