# Peer review of "Whole-Genome Omics Elucidates the Role of CCM1 and Progesterone in Cerebral Cavernous Malformations within CmPn Networks"

_diagnostics, 2024, doi:10.3390/diagnostics14171895_

Round 1

Reviewer 1 Report

Comments and Suggestions for Authors

This is a very comprehensive analysis of proteomic and transcriptomic data from cell lines of various expression levels of the Krit1/CCM1 gene, with and without addition of Progesterone.   

I have only two suggestions for improvement.

1.    In figure 5, the legend states there were 12 Transcription factors that were identified. But only one, BCAP31, is listed in the text.  What are the other 11 TFs that were identified?   These can be listed in the text or in a Table.

2.  Given the complexity of the data and the analyses, I found it difficult to understand the biological insights that were identified in this paper. Many pathways were identified in the different analyses, but the take home lesson is not clear to me.    I recommend a final figure showing a graphical  summary of the main conclusions of the paper.

Comments on the Quality of English Language

in a few places, the word "the" or "a" is missing in front of a noun.  This is a common error that can be easily remedied using a grammar check program.

Reviewer 2 Report

Comments and Suggestions for Authors

Dear Authors,

I read with great interest your manuscript Whole-Genome Omics Elucidates the Role of CCM1 and Progesterone in Cerebral Cavernous Malformations within CmPn Networks

However, there are some things that you need to look over. You have many abbreviations in the text, please insert an abbreviations list at the end of the manuscript. This will greatly improve the readability of the manuscript. At least, explain ‘CCM1’ and ‘CmPn, CmP’ at the first appearance in the text.

In the Discussions section, you need to expand more on the therapeutic potential of your findings, not only in CCM, but also in cervicofacial malformations where surgery has important esthetic and functional consequences. This will put in context your research. Reference this to the work  Vrinceanu D, Dumitru M, Marinescu A, Dorobat B, Palade OD, Manole F, Muresian H, Popa-Cherecheanu M, Ciornei CM. New Insights into Cervicofacial Vascular Anomalies. J Clin Med. 2024 Jun 15;13(12):3515. doi: 10.3390/jcm13123515. PMID: 38930043; PMCID: PMC11205235.

Before the conclusions insert a short paragraph about the limitations of the present study.

Congratulations on your great work!

Looking forward to receiving the improved version of your manuscript.
